# Effects of atmospheric transport and trade on air pollution mortality in China

Hongyan Zhao[1,*], Xin Li[1,*], Qiang Zhang[1], Xujia Jiang[1,2], Jintai Lin[3], Glen G. Peters[4], Meng Li[1], Guannan Geng[1], Bo Zheng[2], Hong Huo[5], Lin Zhang[3], Haikun Wang[6], Steven J. Davis[1,7], and Kebin He[1,2]

[1]Ministry of Education Key Laboratory for Earth System Modelling, Department of Earth System Science, Tsinghua University, Beijing, China;
[2]State Key Joint Laboratory of Environment Simulation and Pollution Control, School of Environment, Tsinghua University, Beijing, China;
[3]Laboratory for Climate and Ocean-Atmosphere Studies, Department of Atmospheric and Oceanic Sciences, School of Physics, Peking University, Beijing, China;
[4]Center for International Climate and Environmental Research—Oslo, N-0318 Oslo, Norway;
[5]Institute of Energy, Environment and Economy, Tsinghua University, Beijing, China;
[6]School of Environment, Nanjing University, Nanjing, China;
[7]Department of Earth System Science, University of California, Irvine, CA 92697, USA

[*] *These authors contributed equally to this work.*

*Correspondence to:* Qiang Zhang (qiangzhang@tsinghua.edu.cn) and Steven J. Davis (sjdavis@uci.edu)

**Abstract.** Air quality is a major environmental concern in China, where premature deaths due to air pollution exceed 1 million people per year in recent years. Here, using a novel coupling of economic, physical and epidemiological models, we estimate the premature mortality related to outdoor $PM_{2.5}$ air pollution in seven regions of China in 2010 and show for the first time how the distribution of these deaths in China is determined by a combination of economic activities and physical transport of pollution in the atmosphere. We find that 33% (338,600 premature deaths) of China's $PM_{2.5}$-related premature mortality in 2010 were caused by pollutants emitted in a different region of the country and transported in the atmosphere, especially from north to south and from east to west. Trade further extended the cross-regional impact; 56% of (568,900 premature deaths) China's $PM_{2.5}$-related premature mortality was related to consumption in another region, including 423,800 (42% of total) and 145,100 (14%) premature deaths from interregional (within China) trade and international trade respectively. Our results indicate that multilateral and multi-stage cooperation under a regional sustainable development framework is in urgent need to mitigate air pollution and related health impacts, and efforts to reduce the health impacts of air pollution in China should be prioritized according to the source and location of emissions, the type and economic value of the emitting activities, and the related patterns of consumption.

# 1 Introduction

Atmospheric pollution is a major environmental problem in China, with substantial adverse health effects (Yang et al. 2013; Apte et al., 2015). Between 2006 and 2012, approximately 1.1 billion people (82% of the nation's population) live in areas where the yearly average mass concentrations of fine particulate matter ($PM_{2.5}$) exceeds 35 μg m$^{-3}$ (Geng et al., 2015)—
above the interim target-1 for annual average exposure set by the World Health Organization (WHO, 2005). In turn, this magnitude of exposure has had large impacts on public health and economic output. In 2010, $PM_{2.5}$ pollution alone was linked to 1.2 million premature deaths in China (Yang et al. 2013), or ~35% of all such deaths worldwide (Apte et al., 2015; Brauer et al., 2016), with associated economic losses equivalent to more than 6% of China's GDP (Matus et al., 2012). The distribution of air pollution and attendant impacts vary across Chinese provinces due to differences in physical geography,
meteorology, population density, level of economic development, production structure, and available technologies (Geng et al., 2015; Jiang et al., 2015; Ma et al., 2014). For example, annual average $PM_{2.5}$ concentrations in northern China are roughly 1.5 times greater than the national average and two times greater than concentrations in southern China (Geng et al., 2015). In light of these differences, the central and local governments have established various goals, strategies and measures for reducing air pollution, with varying degrees of success (Lin et al., 2010).

Effective and efficient control of air pollution relies upon an understanding of the pollution sources and their relative environmental impacts. This has led to an increasing number of studies aimed at attributing pollution to sources at high spatial, temporal, and sectoral resolutions (e.g., Chambliss et al., 2014; Zhang et al., 2015; Turner et al., 2015; Li et al., 2015). An important finding of these studies is that regional air quality is in many cases strongly influenced by pollution produced in other regions and transported in the atmosphere across regional boundaries (e.g., Hu et al., 2015; Li et al., 2015).
For instance, a recent study found that during the month of January 2013-2015, roughly half of the $PM_{2.5}$ present in Beijing and Tianjin (47% and 55%, respectively) was due to emissions produced in other regions (Li et al., 2015). Recent work has further investigated the effect of trade on air pollutant emissions (Guan et al., 2014; Huo et al., 2014; Zhao et al., 2015; Meng et al., 2016) and related impacts on air quality, health, and climate (Takahashi et al., 2014; Jiang et al., 2015; Lin et al., 2014, 2016; Li et al., 2016b; Wang et al., 2017; Zhang et al., 2017). Here, we assess for the first time the health impacts of
trans-boundary $PM_{2.5}$ pollution and trade within China. Our results reveal with greater detail than previously the health impacts of specific economic activities (e.g., the production of raw materials and intermediate goods, production of final goods, and consumption of final goods) by region in China. This information may be used by policymakers in the design and evaluation of control strategies that account for cross-regional pollution.

Our analysis entails a novel coupling of physical, economic, and epidemiological models that use the latest available data,
from 2010. Together, these models allow us to estimate premature deaths in China due to local and trans-boundary anthropogenic $PM_{2.5}$ pollution associated with three different economic activities (production of raw materials/intermediate goods, production of final goods, and consumption of final goods) for each of seven regions (North, Yangtze River Delta, Southeast, Central, Northwest, Southwest, and Northeast China; see Table A1 for region definitions).

## 2 Materials and Methods

In this study, four state-of-the-art models were integrated to analyse the drivers of PM$_{2.5}$-related deaths across seven regions in China. Fig. 1 depicts the integrated assessment framework in 4 steps. Below we describe in details for each step.

### 2.1 Estimation of PM$_{2.5}$-related premature deaths

Satellite-based ground-level PM$_{2.5}$ mass concentrations at a 0.5 °×0.667 °resolution used in this study were derived from our previous work (Geng et al., 2015). It was estimated by using the aerosol optical depth (AOD) derived from satellite instruments and conversion factors between AOD and PM$_{2.5}$, simulated by the GEOS-Chem chemical transport model (Bey et al., 2001). The satellite-based AOD were generated by combining results from MODIS and MISR instruments onboard the Terra satellite, after being filtered by using ground-based AOD measurements. The conversion factors between AOD and

PM$_{2.5}$ were calculated by the nested GEOS−Chem model over China at a resolution of 0.5° × 0.667° (Chen et al., 2009). And for this simulation, anthropogenic emissions over China were taken from the Multi-resolution Emission Inventory of China (MEIC: http://www.meicmodel.org/), a updated version of the technology-based, bottom-up pollution inventory developed by Tsinghua University (Zhang et al. 2009; Lei et al. 2011; Liu et al 2015). Note that, in this study the simulated proportion of mineral dust in surface PM$_{2.5}$ was subtracted to exclude the impact of natural mineral dusts on premature deaths, and we

assumed that the contribution of dust to PM$_{2.5}$ concentrations was proportional to their previously-estimated air pollution disease burden (Chafe et al., 2015; Bhalla et al., 2014).

The Integrated Exposure-Response (IER) model developed by Burnett et al. (2015) is used in this work to describe the concentration−response relationship between long term exposure to PM$_{2.5}$ (annual mean values in this study) and premature deaths for various leading causes. It is fitted by incorporating information from cohort studies of ambient air pollution,

second hand tobacco smoke, household solid, cooking fuel, and active smoking (Burnett et al., 2015). The relative risk (*RR*) was calculated as:

$$RR_l(C) = \begin{cases} 1 + \alpha_l \left( 1 - e^{-\gamma_l (C-C_0)^{\delta_l}} \right), if\ C > C_0 \\ 1 \qquad\qquad\qquad\quad ,else \end{cases} \tag{1}$$

where $C$ is the annual mean PM$_{2.5}$ concentrations in 2010; $C_0$ is the counterfactual concentration; $l$ represents a given health effect; and $\alpha_l$, $\gamma_l$, and $\delta_l$ are parameters used to describe the shape of the concentration-response curve (Burnett et al., 2015).

The RR then was can be converted to the attributable fraction (AF):

$$AF = \frac{RR-1}{RR} \tag{2}$$

The health outcomes or mortality attributable to PM$_{2.5}$ was then estimated:

$$M = AF \times B \times P \tag{3}$$

where B is the death incidence of a given health effect derived from the national average data in GBD2013 (Forouzanfar et al., 2015); P is the size of the exposed population obtained from the LandScan global population database (Bright et al., 2011) .

Following the previous studies (Jiang et al., 2015; Lee et al., 2015), in this work we mainly focus on four leading causes of the PM$_{2.5}$-related premature deaths: ischemic heart disease (IHD), stroke, chronic obstructive pulmonary disease (COPD) and lung cancer (LC). And according to the Global Burden of Disease (GBD) projects (Forouzanfar et al., 2015), we assume that these PM$_{2.5}$-related health impacts are source and composition independent. Data for $C_0$ and B can be found in table A2.

## 2.2 Contribution of individual source emissions to regional premature deaths

Following Lee et.al (2015), the GEOS-Chem adjoint (version 3.5, driven by MEIC inventory) model combined with the IER model were applied over East Asia (11 °S-55 °N, 70 °E-150 °E) at a resolution of 0.5 °×0.667 °to calculate the contributions of location- and species-specific emissions to PM$_{2.5}$-related premature deaths in individual regions (see region definitions in Table A1).

Firstly, we defined eq.3 as the adjoint cost function (or concentration-dependent function) of a given region (e.g. region $r$); and the total value of M$^r$ is with respect to the satellite-based PM$_{2.5}$ in region $r$ obtained from Geng et.al (2015). We then used the adjoint model to calculate partial derivatives of this cost function with respect to anthropogenic emissions of individual species ($\frac{\partial M^r}{\partial E_{i,j,k}}$), which we referred to as the sensitivity of region $r$'s premature deaths (M$^r$) to gridded emissions ($E_{i,j,k}$; $i$ , $j$ and k are indices for longitude, latitude and species) in the simulation domain. A single adjoint simulation provided sensitivities of M$^r$ with respect to emissions at all species, locations and times (Lee et al., 2015; Pappin and Hakami, 2013; Turner et al., 2015). After computing the model sensitivities, we multiplied the emission sensitivity by the amount of emissions to obtain a semi-normalized sensitivity (SS) (Henze et al., 2007; Henze et al., 2009; Turner et al., 2015; Dedoussi and Barrett, 2014), which means the contribution of species- and location-specific emissions to the premature deaths (Turner et al., 2015; Dedoussi and Barrett, 2014):

$$SS^r_{i,j,k} = \frac{\partial M^r}{\partial E_{i,j,k}} \times E_{i,j,k} \tag{4}$$

Then, a normalized *SS* (hereafter P), which represents the percentage contribution of source- specific emissions to premature deaths was calculated as:

$$P^r_{i,j,k} = \frac{SS^r_{i,j,k}}{\sum_i \sum_j \sum_k SS^r_{i,j,k}} \times 100\% \tag{5}$$

The normalization process minimizes the effects of nonlinear relation between emissions and pollutant concentrations and between concentrations and mortality. A similar approach was taken by Li et.al (2016a) for attributing ozone radiative forcing to individual countries.

For this work, we calculated responses to absolute changes in $NH_3$, $SO_2$, $NO_x$, BC, OC and anthropogenic $PM_{2.5}$ dust (Zhang et al., 2015). And a total of seven groups of GEOS-Chem adjoint model simulations were conducted; one group for each receptor region. In order to reduce the computation costs, four months (January, April, July and October of 2010) of simulations were conducted for each group. Results for these four months are averaged to represent the annual mean *SS* in

2010.

## 2.3 Regional pollutant emissions attributed to regions producing final goods and regions consuming the final goods.

The production of a specific product or service represents one stage in a supply chains because such production requires material and energy inputs and may in turn supply other production processes (i.e., the products are intermediate) or final sales (i.e., the products are finished goods ready for final consumption) (Davis et al., 2011). Using the 30-province, 30-sector

Multi-regional Input-Output (MRIO) model of China compiled by Liu et.al (2014), we attribute the emissions released in a region (i.e., the producer) to both final produces in supply chains (who produced the finished products using intermediate inputs made locally or imported from other regions, here we call this regions as "assembler") and final consumers (who ultimately consume the finished products).

The MRIO analysis starts with the monetary flows between sectors and regions:

$$
\quad \begin{pmatrix} \mathbf{x}^1 \\ \mathbf{x}^2 \\ \mathbf{x}^3 \\ \vdots \\ \mathbf{x}^m \end{pmatrix} = \begin{pmatrix} \mathbf{A}^{1,1} & \mathbf{A}^{1,2} & \mathbf{A}^{1,3} & \cdots & \mathbf{A}^{1,m} \\ \mathbf{A}^{2,1} & \mathbf{A}^{22} & \mathbf{A}^{2,3} & \cdots & \mathbf{A}^{2,m} \\ \mathbf{A}^{3,1} & \mathbf{A}^{3,2} & \mathbf{A}^{3,3} & \cdots & \mathbf{A}^{3,m} \\ \vdots & \vdots & \vdots & \ddots & \vdots \\ \mathbf{A}^{m,1} & \mathbf{A}^{m,2} & \mathbf{A}^{m,3} & \cdots & \mathbf{A}^{m,m} \end{pmatrix} \begin{pmatrix} \mathbf{x}^1 \\ \mathbf{x}^2 \\ \mathbf{x}^3 \\ \vdots \\ \mathbf{x}^m \end{pmatrix} + \begin{pmatrix} \sum_r \mathbf{y}^{1,s} \\ \sum_r \mathbf{y}^{2,s} \\ \sum_r \mathbf{y}^{3,s} \\ \vdots \\ \sum_r \mathbf{y}^{m,s} \end{pmatrix} \quad (6)
$$

where $x^r$ is a vector of the total economic output of each sector in province *r*; $y^{r,s}$ is a vector of the finished products by each sector produced in region *r* and consumed in region *s*; $\mathbf{A}^{r,s}$ is a normalized matrix of intermediate coefficients in which the columns reflect the input from the sectors in region *r* required to produce one unit of output from each sector in region *s;* and *m* is the total province number (here *m*=30). Solving for total output, eq can be written as:

$\mathbf{X} = (\mathbf{I} - \mathbf{A})^{-1}\mathbf{y}$  (7)

where $\mathbf{I}$ is identity matrix, $\mathbf{A}$ is the block matrix in eq.6, and $(\mathbf{I-A})^{-1}$ is the Leontief inverse matrix.

Under this framework, pollutant emissions embodied in the trade flow can be calculated as:

$\mathbf{E} = \hat{f}(\mathbf{I} - \mathbf{A})^{-1}\mathbf{y}$  (8)

where $\hat{f}$ is the diagonalization of the vector of region-specific pollutant emissions for unit output of each sector.

Region- and sector-specific emissions attributed to assembler region *s* can be calculated as follows:

$$\mathbf{e}_{asse}^{s} = \hat{\boldsymbol{f}}(\mathbf{I} - \mathbf{A})^{-1} \begin{pmatrix} 0 \\ \vdots \\ \sum_{r} \mathbf{y}^{s,r} \\ \vdots \\ 0 \end{pmatrix} \tag{9}$$

where $\mathbf{e}_{asse}^{s} = (\mathbf{e}_{asse}^{1,s}\ \mathbf{e}_{asse}^{2,s}\mathbf{e}_{asse}^{3,s}\ ...\ \mathbf{e}_{asse}^{m,s})$'; and $\mathbf{e}_{asse}^{r,s}$ is a sector-specific vector for emissions occurred in region $r$ caused by producing intermediate products to be assembled (as finished products) in region $s$.

Region- and sector-specific emissions attributed to consumer region $s$ can be calculated as:

$$5 \quad \mathbf{e}_{cons}^{s} = \hat{\boldsymbol{f}}(\mathbf{I} - \mathbf{A})^{-1} \begin{pmatrix} \mathbf{y}^{1,s} \\ \mathbf{y}^{2,s} \\ \mathbf{y}^{3,s} \\ \vdots \\ \mathbf{y}^{m,s} \end{pmatrix} \tag{10}$$

where $\mathbf{e}_{cons}^{s} = (\mathbf{e}_{cons}^{1,s}\ \mathbf{e}_{cons}^{2,s}\mathbf{e}_{cons}^{3,s}\ ...\ \mathbf{e}_{cons}^{m,s})$'; and $\mathbf{e}_{cons}^{r,s}$ is a sector-specific vector for emissions occurred in region $r$ caused by production of intermediate or final products to be consumed in region $s$.

In this section, sector specific emissions to produce $\boldsymbol{f}$ in eq.8 to eq.10 were derived from mapping process between MEIC model and sectors defined in the MRIO model for each provinces, which can be found in our previous studies (Huo et al., 2014; Zhao et al., 2015). Within each region, emissions attributed to each activator (assembler or consumer) can be allocated to individual locations (grid cells) based on the sector-spatial distribution in MEIC and the attributed ratios:

$$R_{k,asse}^{r,s} = \mathbf{e}_{k,asse}^{r,s}/\mathbf{e}_{k}^{r} \tag{11}$$

$$R_{k,cons}^{r,s} = \mathbf{e}_{k,cons}^{r,s}/\mathbf{e}_{k}^{r} \tag{12}$$

where $\mathbf{e}_{k}^{r}$ is sector-specific emissions vector (species $k$) produced in region $r$; $R_{k,asse}^{r,s}$ and $R_{k,cons}^{r,s}$ are sector-specific ratios of emissions occurred in region $r$ but allocated to region $s$ from assembler and consumer perspectives, respectively. As part of our calculation, we aggregated the interregional emission impact of 30 provinces into 7 regions, as defined in Table A1.

## 2.4 Premature deaths attributed to regions producing final goods and regions consuming the final goods

Results from above three steps were integrated to attribute regional- and source-specific PM$_{2.5}$ deaths to specific economic activities (i.e. the production of final goods by the "assembler", and the ultimate consumption of those goods) in specific regions along supply chains as:

$$M_{asse}^{s} = \sum_{r} \mathrm{M}^{r} \sum_{t} \sum_{k} (P_{(i,j)\in t,k}^{r} \times R_{(i,j)\in t,k,asse}^{t,s}) \tag{13}$$

$$M_{cons}^{s} = \sum_{r} \mathrm{M}^{r} \sum_{t} \sum_{k} (P_{(i,j)\in t,k}^{r} \times R_{(i,j)\in t,k,cons}^{t,s}) \tag{14}$$

where $M_{asse}^s$ and $M_{cons}^s$ mean premature deaths attributed to region $s$ from assembler and consumer perspectives, respectively; $R_{(i,j) \in t,k,asse}^{t,s}$ and $R_{(i,j) \in t,k,cons}^{t,s}$ are sector average ratios of emission occurred in grid $(i,j)$ relocated to region s from assembler and consumer perspectives, respectively.

## 3 Results

### 3.1 National and regional mortality attributed to anthropogenic PM$_{2.5}$

In 2010, China's population-weighted PM$_{2.5}$ concentrations caused by anthropogenic emissions reached 53 $\mu g$ m$^{-3}$, leading to 1.02 (95% CI: 0.64-1.22) million premature deaths, which accounted for about 35% of the global total mortality from ambient PM$_{2.5}$ (Apte et al., 2015). Adding another 0.23 (95% CI: 0.14-0.27) million premature deaths from windblown natural dusts, our estimate of premature deaths is within 2% of the result of GBD 2010 for China (1.27 million premature deaths; Brauer et al., 2015).

Table 1 and Fig. 2A show in detail about regional anthropogenic PM$_{2.5}$ concentrations and related mortality. As figures and table shown, atmosphere pollution and related health impact vary substantially across the seven Chinese regions. Dominated by heavy industries, North region suffered the most severe pollution, and its population-weighted mean PM$_{2.5}$ concentrations reached 82 $\mu g$ m$^{-3}$, followed by the Central (67 $\mu g$ m$^{-3}$), Southwest (52 $\mu g$ m$^{-3}$) and Yangtze River Delta (50 $\mu g$ m$^{-3}$). However, considering the total population exposed to pollution, the Central region had the highest mortality (302,200 premature deaths; 95%CI: 187,400-359,900) and a high mortality ratio (90 deaths per 10$^5$ people; 95%CI: 56-107), followed by the Southwest (195,200 premature deaths; 95%CI: 121,900-234,600) and the North (182,200 premature deaths; 95%CI: 115,700-213,600).

### 3.2 Effects of atmospheric transport of air pollution on regional mortality

Regional atmospheric pollution and related health impacts can be attributed to emissions from both local and other regions as a result of atmospheric transport. Further, emissions in a given region can also be attributed to regions who consuming the related products due to trade, thus pollution induced mortality can finally be attributed to the consuming regions. Table 2 (and Fig. 2B and 2C) presented the source attribution of regional PM$_{2.5}$ exposure and related premature deaths from both production and consumption perspectives.

As shown in Table 2 and Fig. 2B, in the year 2010, 33% of total premature deaths due to outdoor PM$_{2.5}$ exposure were caused by trans-boundary pollution, and ratios for specific regions vary from 30% in Northeast to 40% in Northwest. Among these, less than 1% was caused by pollution transported from region out of China. Fig. 3A further shows the effect of atmospheric transport on premature deaths in each Chinese region due to PM$_{2.5}$ air pollution produced in other regions, with particularly large interregional impacts highlighted by arrows. The red shading in Fig. 3A corresponds to regions (e.g., the North, Yangtze River Delta and Northwest) whose emissions caused a greater number of deaths in other regions than

pollution in other regions caused in that region—a net export of premature mortality. In contrast, the blue shaded regions (the Southwest, Southeast and Central) experienced greater numbers of deaths due to extra-regional emissions than their emissions caused in other regions. Regionally, pollution from the North region that was transported in the atmosphere to the populous Central and Yangtze River Delta regions is particularly harmful and causes the most premature deaths, with 38,100

(95% CI: 23,600-45,400) and 18,200 (95% CI: 11,200-22,300) premature deaths related to these trans-boundary flows, respectively. Perhaps due to its substantial emissions and central location in the country, premature deaths occurred in the Central region by emissions produced elsewhere (101,400; 95% CI: 62,900-120,700) and deaths caused by Central region's emissions transported elsewhere (91,900; 95% CI: 57,500-110,400) are approximately equal, and Southwest endured the most premature deaths from emission in Central. Nationally, the net flows of trans-boundary $PM_{2.5}$-related health impact

mainly caused by pollution transported from north to south and from east to west.

### 3.3 Effects of interregional trade on regional mortality

Compared to the physical atmospheric transport, trade enable the cross-regional impact more broadly (Table 2 and Fig. 2C), as the production of emissions can occurred far from where the products were finally consumed. Nationally, 56% of $PM_{2.5}$-related premature mortality in China in 2010 was linked to consumption in a different region through both interregional

(within China) and international trade activities. Among these, 42% of premature mortality was associated with interregional trade, and the ratios vary from 38% in Northeast to 49% in Northwest. International export accounted for approximately 14% of total $PM_{2.5}$-related premature mortality in China in 2010, comparable to 12% in 2007 reported by Jiang et al. (2015).

For a finished product or service, it may experience different stages before being sold to final consumers, such as material production and products assemble, these may occurred in different regions. Fig. 3B shows the effect of trade between the

region producing a raw material or intermediate good and the region producing the final good ready for consumption. This is important because in many cases the region assembling or otherwise preparing the final good is able to capture a large fraction of the final good's value without undertaking more energy- and pollution-intensive processes that were required to produce the raw materials and intermediate goods (Prell et al., 2014; Liu et al., 2016). This distinction is particularly relevant in China because previous studies have shown that more affluent coastal provinces in China are increasingly importing

intermediate goods and materials from less-developed provinces (Feng et al., 2013; Jiang et al., 2015; Zhao et al., 2015). Here, we find that premature deaths related to final goods produced in red-shaded regions are substantially greater than the deaths due to the emissions produced in those regions. In particular, final goods assembled/manufactured in the Yangtze River Delta and Southeast regions led to 56,200 (95% CI: 35,100-67,300) and 33,600 (95%CI: 20,900-40,200) deaths due to emissions in other regions, respectively. In contrast, blue-shaded regions like the Central, Northwest and Southwest are those

which disproportionately produce and export raw materials and intermediate goods (e.g., mineral ores and metals), and therefore suffer health impacts to support the manufacture of final goods in other regions. For example, 15% of deaths caused by emissions in the Central region are related to final goods manufactured in the North, Yangtze River Delta, and Southeast regions.

With supply chains or trade extending, finished products may finally be consumed by another region. Fig. 3C further shows the full effect of trade from the producer of emissions related deaths to the final consumer in map. As this study doesn't include premature deaths caused by international imports, in this figure we only present regional production related deaths caused by domestic consumption, premature deaths induced by goods and services produced in China for international export are shown in Fig. 4 separately. As Fig. 3C shows, deaths related to consumption in red-shaded regions are substantially greater than the deaths number caused by emissions produced in those regions. Note that, in this figure Central region shows net export of production related premature deaths with all other six regions, this can be attributed to its abundance interregional export, severe pollution and high population density. Moreover, Fig. 3C also highlights the case of Northeast. Even though Northwest shows net pollution export with other regions (Zhao et al., 2015), but its exported emissions cause less deaths than the relative small emissions occurred on other regions to support consumption in Northeast, just because that population and production intensities in Northeast are far less than those in other regions, such as Central and North regions.

Fig. 3D shows the combined effect of atmospheric transport and trade on each region. Same as Fig. 3C, deaths caused by international export were excluded in this map. Here, premature deaths related to final goods consumed in red-shaded regions are substantially greater than the deaths occurred in those regions. Even though similar with Fig. 3C, Fig. 3D highlight that atmospheric transport aggravated the premature deaths transferred from North and Y.R.D to Central region.

Fig. 4 attributed China's PM$_{2.5}$-related premature deaths embodied in international export to seven Chinese regions where the emissions were produced and where the final products were exported. As shown, of these international exports, roughly three-quarters (76%) of the related deaths are associated with the exports from the east coast regions (North, Yangtze River Delta and Southeast). However, only 59% of deaths related to exports from these coastal regions are caused by emissions actually produced in those regions, and even fewer (49%) of associated deaths actually occurred in those regions (Fig. 2B). These results emphasize that international exports commonly entail intermediate inputs from less-developed regions of China (e.g., the Central region; Feng et al., 2013; Zhao et al., 2015).

Summing from the previous sections, Fig. 5 integrates the results related to both atmospheric transport and trade to show PM$_{2.5}$-related premature mortality in each region due to local and other regions' manufacturing and consumption activities, and separating the effects of locally-produced and atmospherically-transported pollution. For a given region, emissions produced in the region to supply either local or other regions' consumption accounted for the largest share of deaths in the region (60-70%; purple and light blue bars in Fig. 5A), followed by the 'spillover impact' of emissions produced in other regions that are not related to the local region's manufacturing or consumption activities (27-37%; gray bars in Fig. 5A). Emissions produced in other regions and related to the local region's consumption contributed 1-3% of each regions' mortality, via atmospheric trans-boundary transport (dark blue bars in Fig. 5A). Finally, the effect of atmospheric transport from other countries contributed only 1-2% of deaths in any Chinese region (light purple bars in Fig. 5A).

Fig. 5B further breakdown the regions involved in each region's spillover impacts according to where the emissions were produced. The magnitude of spillover deaths depends largely on a given region's population and the extent of emissions in their upwind regions. For instance, 75% and 66% of spillover deaths in the Yangtze River Delta and Southwest regions are linked to emissions in upwind regions (primarily the Central and North regions), respectively. As the most populated region, the Central region suffered the most spillover deaths (96,100; 95% CI: 59,600-114,400), 69% of which were related to emission produced in the North and Yangtze River Delta regions.

## 3.4 Uncertainties and limitations

The calculation of premature deaths caused by atmospheric transport and trade is subject to a number of uncertainties and limitations. Bottom-up emission inventories are uncertain due to incomplete knowledge of activity, technology distribution and emission factors. The uncertainties in China's emission inventory were estimated to be −14%~13%, −13%~37%,−17%~54%, −25%~136% and −40%~121% for $SO_2$, $NO_x$, $PM_{2.5}$, black carbon (BC), and organic carbon (OC), respectively (Zhao et al., 2011). Although the quantitative uncertainties are not provided by the MEIC inventory, it has been widely used in chemical transport models and validated against surface and satellite observations (e.g., Chen et al., 2015; Geng et al., 2015; Li et al., 2015; Zheng et al., 2015; Hu et al., 2016).

Uncertainties from the simulation of GEOS-Chem and its adjoint subject to their limitations or errors in chemical and physical representation, such as the chemical conversion, diffusion, deposition, and advection transport. Here we conduct a comparison of the modelled and the satellite-derived $PM_{2.5}$ concentration, and use the normalized mean error (NME) between these two datasets over China seven regions to represent the overall model errors. As shown in Fig. 6, the NME various among seven regions, and range from the lowest in North (30%) to the highest in Northwest (71%). Note that the two datasets agree reasonably well, the R range from 0.67 to 0.95 for seven regions. This provides confidence that the results of this study are based on realistic simulation. The adjoint model may introduce additional uncertainties due to lack consideration of the nonlinear response of the predicted concentration to perturbation of emission input. However, due to its complex in backward calculation and integration with the forward model, there are very few statistical quantification for its uncertainties so far. Lee et.al (2015) used ±40% to represent the total uncertainties caused by the GEOS-Chem adjoint model.

Uncertainties in satellite-derived $PM_{2.5}$ map is ±5% on average according to GBD 2013 (Brauer et al., 2016), as it has been calibrated by satellite-based and surface observations. Uncertainties in IER model are relatively high, mainly arising from the model itself, as it is fitted by limited information on actual exposure to $PM_{2.5}$ for source-specific relative risks. Burnett et.al (Burnett et al., 2015) estimated the uncertainties from IER model by using simulation approach, and they fitted out 1000 sets parameters for the IER function to represent its possible shape. Additionally, the IER model is limited to several assumptions, e.g. $PM_{2.5}$-related health impact is independent of exposure period, $PM_{2.5}$ composition and toxicity for particles from different sources (Burnett et al., 2015; Jiang et al., 2015; Lee et al., 2015).

Additional uncertainties originate from MRIO analysis when linking trade among different regions. MRIO model inherit all uncertainties in its source (survey) data and data manipulation (Peters, 2007; Weber, 2008; Wiedmann et al., 2011; Wiedmann, 2009). In addition, MRIO analysis is limited to sector detail, region coverage, and the number of environmental extensions (Tukker and Dietzenbacher, 2013). Moreover, the China domestic MRIO model, which consider no effect from international import (Hummels et al., 2001), can also introduce some uncertainties. The study conducted by Lin et al (2014) concluded that the uncertainties in Chinese input-output model contributed to ~10% of total errors in export-related pollutant emissions.

A comprehensive uncertainty analysis combining all affecting factors above is difficult due to the limitations of the computational loads. The uncertainty ranges presented in previous sections only represent the uncertainties in IER function, which is obtained by 1000 sets runs of IER parameters fitted by Burnett et.al (2015) to calculate the possible distribution of regional premature mortality.

## 4 Discussion and Conclusions

Patterns of atmospheric $PM_{2.5}$ pollution and resulting premature deaths in China are the result of complex interacting physical transport processes and economic activities (Lin et al., 2014). We found that, in 2010, about one third of $PM_{2.5}$-related premature mortality in China was caused by regional air pollution transport. In the meanwhile, large numbers of premature deaths are caused by economic activities in a different region from where the deaths occurred. For the year 2010, 42% and 14% of $PM_{2.5}$-related premature deaths were associated with interregional and international trade respectively. More economically-developed regions (e.g., the Yangtze River Delta, Southeast and North regions) tend to externalize their emissions and related health impacts by importing goods from less economically-developed regions (e.g., the Central region) , and the frequent wind from north to south and from east to west further aggravated the cross-regional impacts. Thus relocating emissions within the nation will not completely alleviate the environmental and health burden; atmospheric transport of pollution often leads to health impacts in downwind regions. To reduce pollution and relative health impacts effectively, regions should promote interregional technology cooperation including both production and emission control technologies. Further, as main final consumers, the east coast regions can lead a "greening supply chains" action by import more green products, thus exerting a cleaning effect on its upstream production chains (Skelton, 2013).

As a main driver of China's production, international export accounted for 14% of China's anthropogenic $PM_{2.5}$ -related premature deaths in 2010. Moreover, its impact was not evenly distributed among regions, as the developed east coastal regions partly transfer their export related premature mortality to the less developed central and west regions by importing raw material from the less developed central and west regions (Fig. 3B and Fig. 4). This exerts disproportionally life loss and economic gains from international exports among regions (Jiang et al., 2015). Besides, the added pollution results from international export can further affect other countries atmospheric environment (Lin et al., 2014; Lin et al., 2016) or even

premature deaths (Zhang et al., 2017) through cross-continental atmospheric transport. Thus a jointed pollution mitigation action among regions, nations and even production chains is in urgent needed, not only for domestic equality in development, but also for global human health.

Our results represent the most detailed analysis of air pollution mortality in China, its sources, and its underlying economic drivers. Based on these findings, future measures to alleviate these health impacts could be prioritized according to the source and location of emissions as well as the type and economic value of the emitting activities and related patterns of consumption.

*Acknowledgements.* This study was supported by the National Science Foundation of China (41625020, 41629051, 41541039 and 41222036) and China's National Basic Research Program (2014CB441301). G.G. Peters was funded by the Norwegian Research Council (235523). Q. Zhang and K. B. He are supported by the Collaborative Innovation Center for Regional Environmental Quality.

## Appendix A

**Table A1. Region Definitions**

| Region | Provinces/municipalities included in each region |
|---|---|
| **North** | Beijing, Tianjin, Hebei and Shandong |
| **Yangtze River Delta** | Shanghai, Jiangsu and Zhejiang |
| **Southeast** | Fujian, Guangdong and Hainan |
| **Central** | Henan, Anhui, Hubei, Hunan and Jiangxi |
| **Northwest** | Shaanxi, Shanxi, Gansu, Qinghai, Ningxia, Xinjiang and Inner Mongolia |
| **Southwest** | Sichuan, Chongqing, Guizhou, Yunnan and Guangxi |
| **Northeast** | Liaoning, Jilin and Heilongjiang |

**Table A2. Counterfactual concentrations and deaths incidences used in IER model**

| | IHD | Stroke | COPD | LC | Source |
|---|---|---|---|---|---|
| $C_0$ | 6.96 | 8.38 | 7.17 | 7.24 | Lee et.al., 2015 |
| $B$ | 0.000707 | 0.00129 | 0.000696 | 0.000383 | Forouzanfar et al., 2015 |

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

**Table 1.** Regional populations, PM$_{2.5}$ concentrations, mortality and mortality ratios within China

| Region | Population (millions) | Population weighted PM$_{2.5}$ concentrations ($\mu g$ m$^{-3}$) | Mortality (thousands of deaths) | Mortality ratio (deaths per 10$^5$ persons) |
|---|---|---|---|---|
| North | 192 | 82 | 182(116-214, 95%CI) | 95 (60-111,95%CI) |
| Yangtze River Delta | 144 | 50 | 116 (76-153) | 81 (50-99) |
| Southeast | 143 | 27 | 78 (52-98) | 55 (35-67) |
| Central | 337 | 67 | 302 (209-415) | 90 (56-107) |
| Northwest | 157 | 34 | 80 (62-118) | 51 (31-61) |
| Southwest | 254 | 52 | 195(132-254) | 77 (48-92) |
| Northeast | 114 | 28 | 65 (42-82) | 57 (35-70) |
| National average/ total | 1354 | 53 | 1018 (636-1222) | 76 (47-91) |

**Table 2.** Regional population-weighted mean PM$_{2.5}$ concentration in 2010 and related premature deaths from production and consumption perspectives. Each number in the cell shows the population-weighted mean PM$_{2.5}$ concentration or premature deaths in the region indicated by the column due to pollution emitted or goods consumed in the region indicated by the row. Numbers in the parentheses presents the fraction of population-weighted mean PM$_{2.5}$ concentration or premature deaths.

| Region | | North | Y.R.D | Southeast | Central | Northwest | Southwest | Northeast |
|---|---|---|---|---|---|---|---|---|
| population-weighted mean PM$_{2.5}$ concentration (µg m$^{-3}$) | | 81.6 | 50.2 | 27.4 | 67.0 | 34.0 | 51.9 | 27.6 |
| **Region where pollution emitted** | North | 57.1(70%) | 8.9(17.6%) | 1.4(5%) | 9.4(14%) | 2.4(7.1%) | 1.4(2.7%) | 4.1(14.7%) |
| | Y.R.D | 5.3(6.5%) | 30.1(60%) | 2.4(8.8%) | 6.2(9.2%) | 0.3(0.9%) | 0.5(1%) | 0.6(2.1%) |
| | Southeast | 0(0.1%) | 0.6(1.2%) | 18.7(68.4%) | 0.8(1.2%) | 0(0%) | 0.7(1.3%) | 0(0%) |
| | Central | 10.9(13.4%) | 6.2(12.4%) | 3.3(11.9%) | 44.8(66.8%) | 5.8(17.1%) | 6.4(12.3%) | 0.7(2.5%) |
| | Northwest | 5.5(6.8%) | 2.6(5.1%) | 0.7(2.4%) | 4(6%) | 21.8(64.2%) | 2.7(5.3%) | 2.2(8%) |
| | Southwest | 0.2(0.3%) | 0.2(0.4%) | 0.5(1.8%) | 1.2(1.8%) | 3.2(9.5%) | 40(76.9%) | 0(0.1%) |
| | Northeast | 2(2.5%) | 1.2(2.4%) | 0.2(0.8%) | 0.5(0.8%) | 0.2(0.7%) | 0.1(0.1%) | 19.6(70.9%) |
| | Out of China | 0.4(0.5%) | 0.5(1%) | 0.2(0.9%) | 0.2(0.3%) | 0.2(0.5%) | 0.2(0.3%) | 0.5(1.7%) |
| **Region where goods consumed** | North | 35.5(43.5%) | 7.4(14.7%) | 1.6(5.8%) | 8.5(12.7%) | 3.8(11.2%) | 2.9(5.5%) | 3.8(13.8%) |
| | Y.R.D | 9(11%) | 17.8(35.5%) | 2.5(9.3%) | 7.6(11.4%) | 2.3(6.8%) | 2.1(4.1%) | 1.7(6.3%) |
| | Southeast | 1.9(2.3%) | 1.6(3.1%) | 10.4(38.2%) | 2.6(3.9%) | 1(3%) | 2.4(4.5%) | 0.4(1.6%) |
| | Central | 9.5(11.6%) | 5.6(11.1%) | 3.1(11.3%) | 30(44.8%) | 4.8(14.1%) | 6.1(11.7%) | 1.3(4.6%) |
| | Northwest | 6.8(8.4%) | 3.4(6.8%) | 1.2(4.5%) | 4.9(7.4%) | 14.1(41.5%) | 3.2(6.2%) | 2.2(7.9%) |
| | Southwest | 2.1(2.5%) | 1.2(2.4%) | 1.3(4.6%) | 2.8(4.2%) | 3.2(9.5%) | 29.7(57.1%) | 0.4(1.6%) |
| | Northeast | 4.3(5.3%) | 2(4.1%) | 0.5(1.8%) | 1.8(2.7%) | 1.1(3.2%) | 0.7(1.3%) | 13.9(50.2%) |
| | Out of China | 12.1(14.8%) | 10.7(21.4%) | 6.5(23.7%) | 8.5(12.7%) | 3.5(10.2%) | 4.8(9.2%) | 3.4(12.3%) |
| **Premature mortality (100 person)** | | 1822 | 1160 | 785 | 3022 | 796 | 1952 | 649 |
| **Region where pollution emitted** | North | 1258(69.1%) | 182(15.7%) | 44(5.7%) | 381(12.6%) | 64(8.1%) | 69(3.6%) | 97(15%) |
| | Y.R.D | 130(7.1%) | 727(62.7%) | 79(10.1%) | 315(10.4%) | 8(1%) | 33(1.7%) | 14(2.2%) |
| | Southeast | 1(0.1%) | 24(2.1%) | 516(65.8%) | 63(2.1%) | 0(0%) | 49(2.5%) | 0(0%) |
| | Central | 202(11.1%) | 135(11.6%) | 93(11.8%) | 2008(66.5%) | 130(16.3%) | 343(17.6%) | 17(2.6%) |
| | Northwest | 135(7.4%) | 44(3.8%) | 21(2.7%) | 145(4.8%) | 480(60.3%) | 82(4.2%) | 54(8.4%) |
| | Southwest | 5(0.3%) | 5(0.5%) | 15(2%) | 76(2.5%) | 91(11.4%) | 1356(69.5%) | 1(0.1%) |
| | Northeast | 74(4.1%) | 30(2.6%) | 7(0.9%) | 24(0.8%) | 16(2%) | 4(0.2%) | 453(69.8%) |
| | Out of China | 17(0.9%) | 13(1.2%) | 8(1.1%) | 9(0.3%) | 7(0.8%) | 15(0.8%) | 13(2%) |
| **Region where goods consumed** | North | 777(42.6%) | 155(13.4%) | 48(6.2%) | 355(11.8%) | 84(10.6%) | 112(5.8%) | 91(14%) |
| | Y.R.D | 204(11.2%) | 426(36.7%) | 78(9.9%) | 360(11.9%) | 53(6.6%) | 93(4.8%) | 42(6.4%) |
| | Southeast | 41(2.2%) | 43(3.7%) | 294(37.5%) | 137(4.5%) | 22(2.8%) | 109(5.6%) | 10(1.6%) |
| | Central | 188(10.3%) | 124(10.6%) | 88(11.2%) | 1349(44.7%) | 109(13.7%) | 294(15.1%) | 31(4.7%) |
| | Northwest | 158(8.7%) | 71(6.1%) | 36(4.6%) | 202(6.7%) | 326(40.9%) | 109(5.6%) | 52(8.1%) |
| | Southwest | 44(2.4%) | 28(2.4%) | 35(4.5%) | 141(4.7%) | 85(10.7%) | 1001(51.3%) | 10(1.6%) |
| | Northeast | 116(6.4%) | 47(4%) | 15(1.9%) | 78(2.6%) | 31(3.8%) | 26(1.3%) | 322(49.6%) |
| | Out of China | 277(15.2%) | 253(21.8%) | 182(23.2%) | 389(12.9%) | 79(10%) | 193(9.9%) | 77(11.9%) |

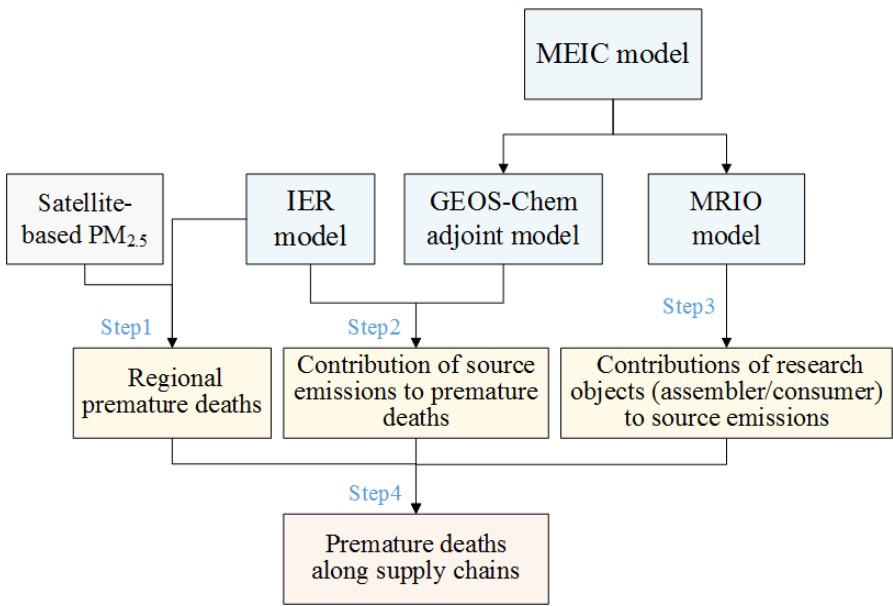

**Figure 1.** Schematic of the methodology used in this study.

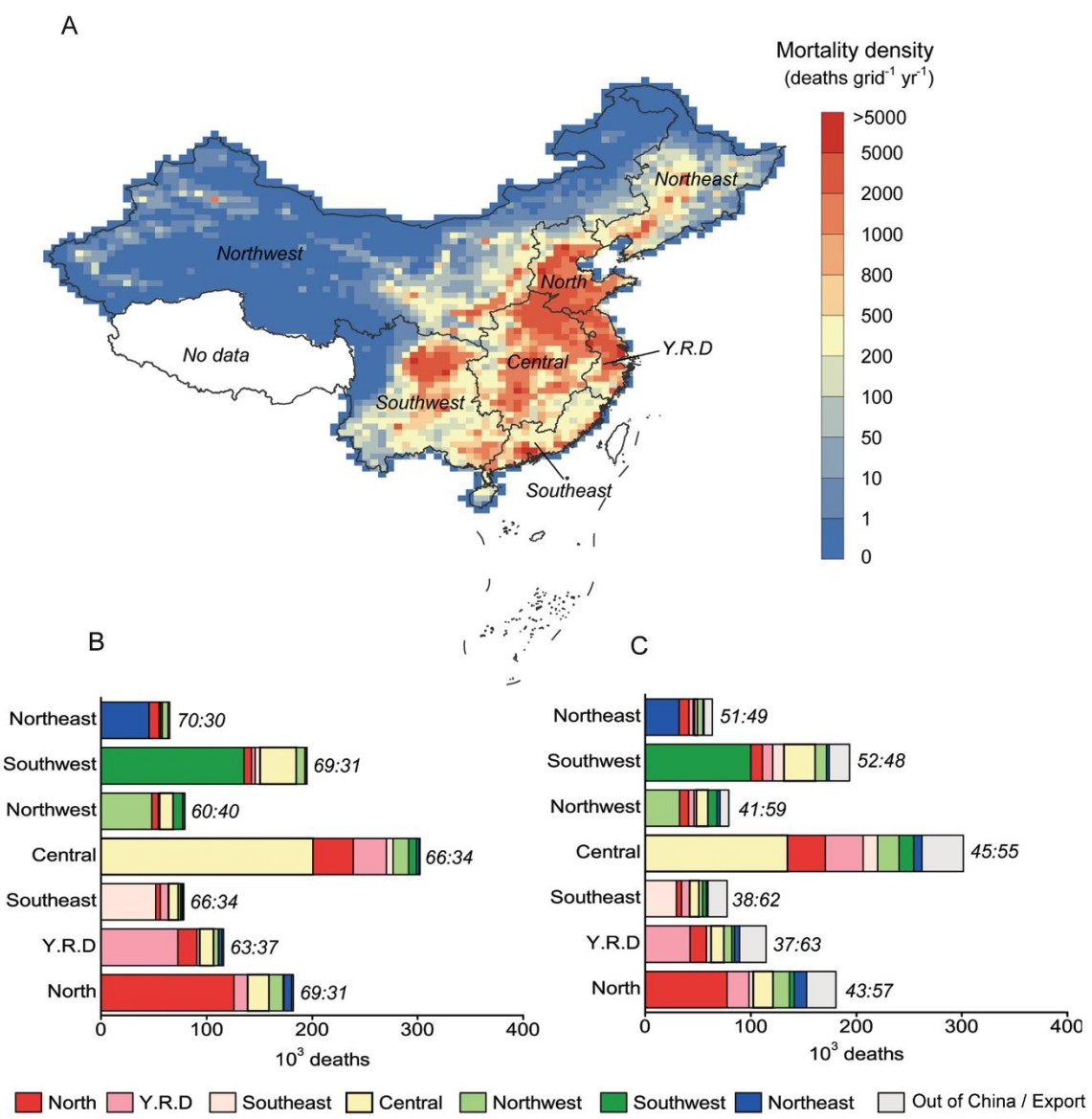

**Figure 2.** Anthropogenic PM$_{2.5}$-related premature deaths in China for 2010 at the 0.5 °×0.667 ° horizontal resolution (A), and regional premature deaths attributed to regions where emissions were produced (B) and regions where products were ultimately consumed (C). Data sets at the end of each bar mean the percentages of regional premature mortality attributed to local source and the percentages attributed to other regions .Y.R.D. is the Yangtze River Delta.

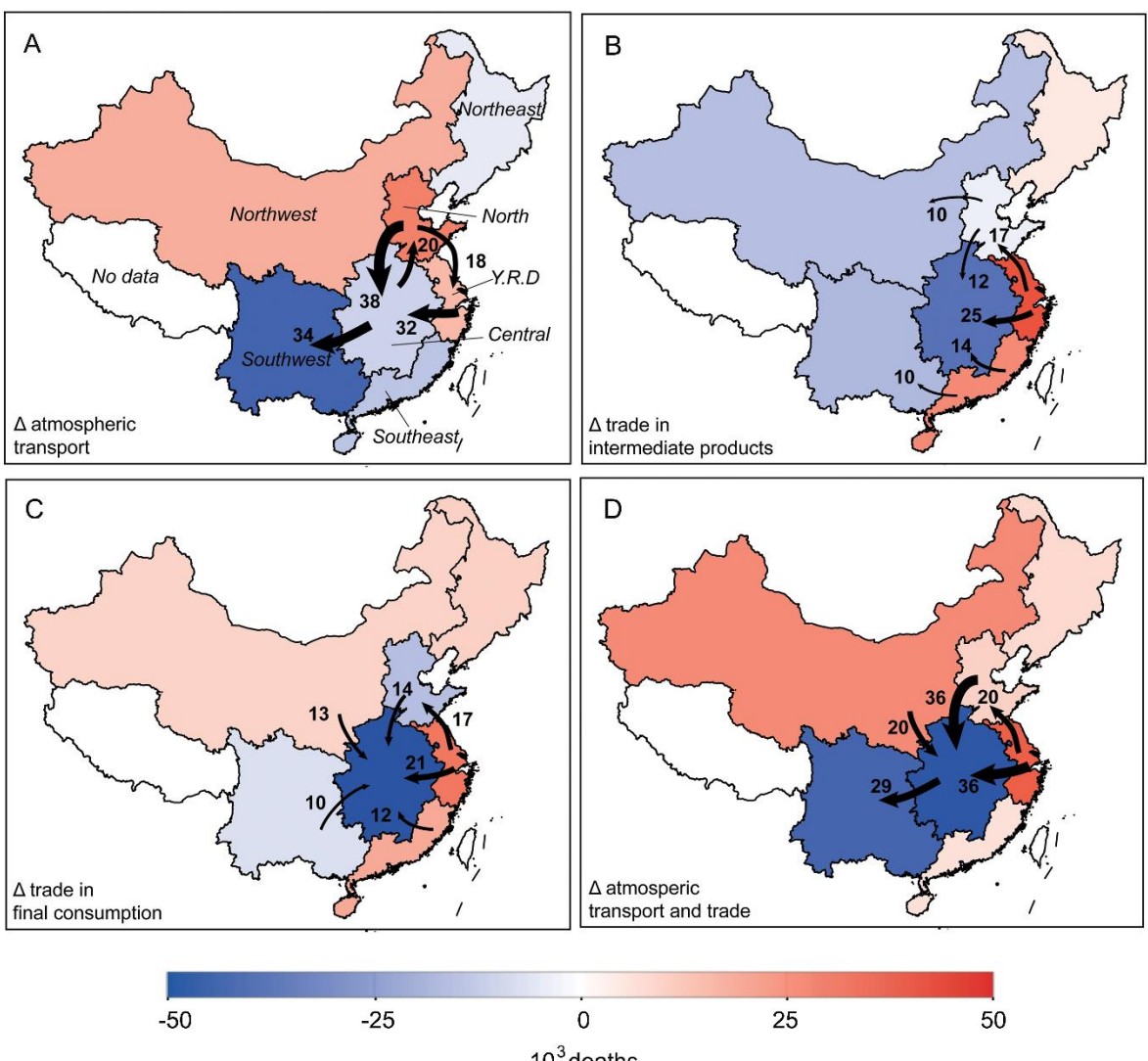

**Figure 3.** The effect of atmospheric transport (A) and trade (B-D) on each region's PM$_{2.5}$-related premature deaths. Panel A compares the number of premature mortality related to emissions produced in each region and deaths that occurred in that region. Panel B and C compare regional production related premature deaths with deaths related to production of final goods in that region, and deaths related to consumption of goods and services in that region, respectively. Panel D compares the number of premature deaths occurred in each region with deaths related to consumption of goods and services in that region. Deaths in other regions due to Chinese pollution and deaths due to emissions in other nations are not included in any of the maps, and international export on regional premature deaths are not included in map C and D. Arrows between regions denote the largest interregional transfers, with numbers of displaced premature deaths shown in thousands.

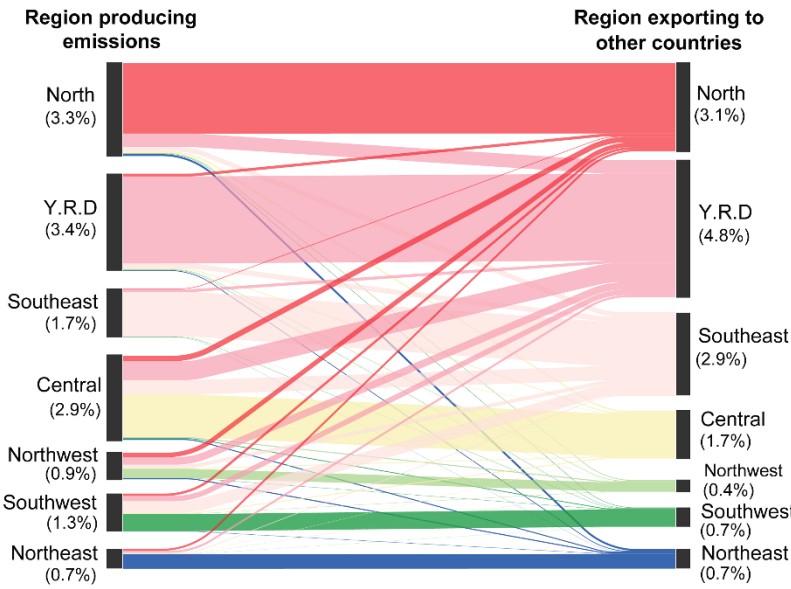

**Figure 4.** Flow map of premature deaths connecting producers and international exporter of finished products. The percentage values are relative to national total anthropogenic $PM_{2.5}$-related premature deaths (1.02 millions).

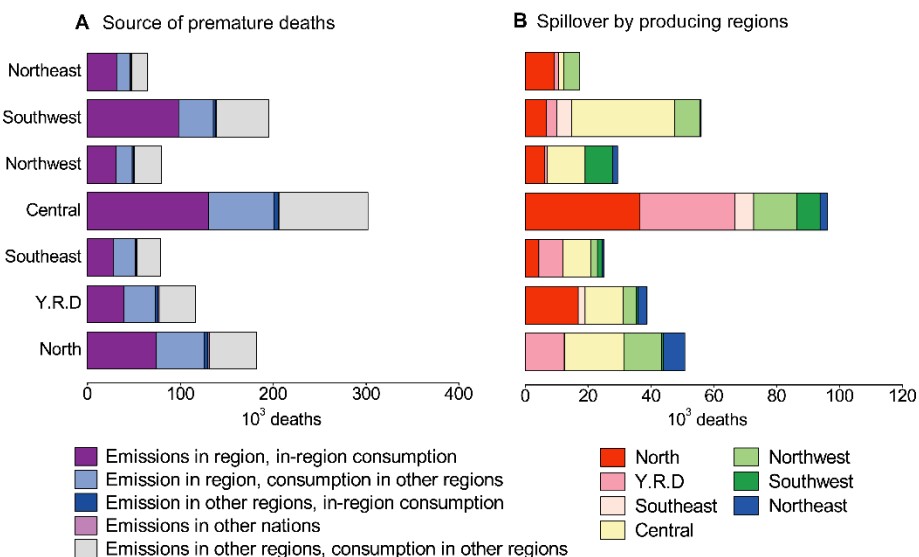

**Figure 5.** Source of PM$_{2.5}$-related premature mortality in each region (A) and their "spillover" source by producing regions (i.e. the gray bars in A: deaths due to emissions in other regions related to goods and services consumed in other regions).

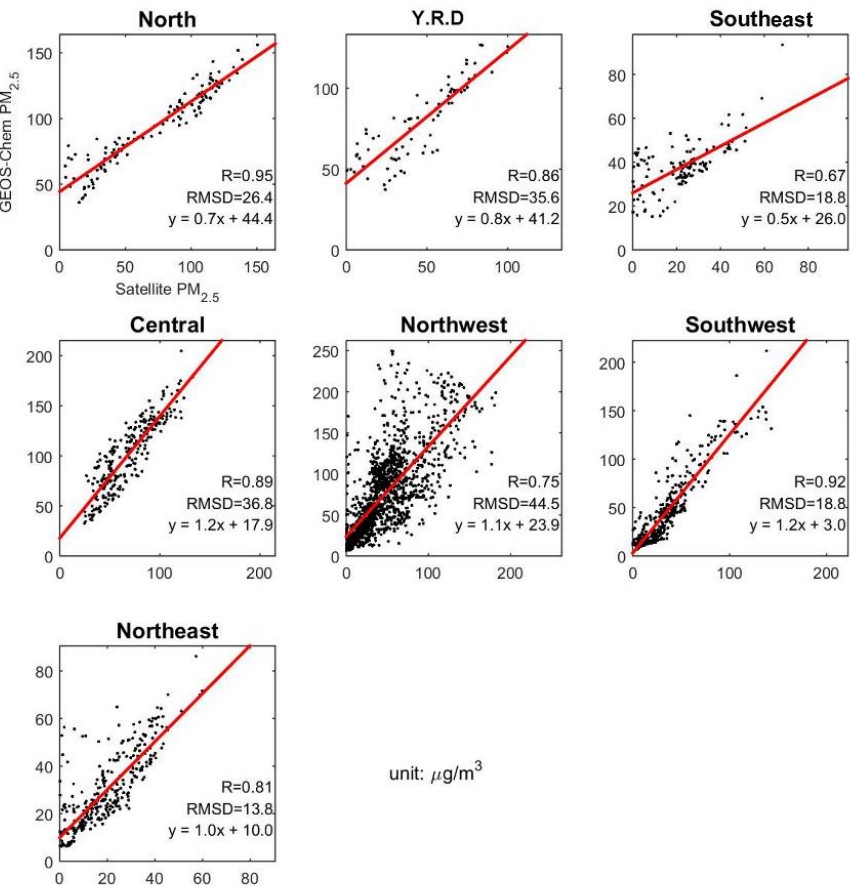

**Figure 6.** Comparisons between the simulated and satellite-derived PM$_{2.5}$ concentrations over the seven China regions.

