# Peer review of "Effects of atmospheric transport and trade on air pollution mortality in China"

_Atmospheric Chemistry and Physics, 2017_

## Referee Comment (RC1) · Anonymous Referee #1 · 5 Apr 2017

This manuscript investigates the effect of atmospheric transport and trade on air pollution mortality in China. The topic is interesting and important. The approach is new and robust. The paper presents a good interdisciplinary study and is written. A few minor correction may need to be made.

1. Abstract: "33% of China's air pollution deaths. . .", it is suggested to provide the number of premature deaths besides the percentage.

2. $C_0$ is the counterfactual concentration in equation (1). The authors should declare the value of $C_0$ adopted in this study.

3. China also imports intermediate and finished goods from other regions. The authors should describe the calculation of international imports in methods.

4. Recent progress in transboundary air pollutants and its associated health effect should be mentioned in this study, some are listed below: Zhang Q, Jiang X, Tong D, et al. Transboundary health impacts of transported global air pollution and international trade[J]. Nature, 2017, 543(7647): 705-709. Li Y, Meng J, Liu J, et al. Interprovincial reliance for improving air quality in China: a case study on black carbon aerosol[J]. Environmental science & technology, 2016, 50(7): 4118-4126.

5, The authors should thoroughly check the manuscripts to avoid grammar mistakes. For example, Page

6 Line 25, "which account for about 35%..." should be "which accounted for about 35%...", all "PM2.5" should be changed to "PM2.5".

6, Fig.2, white color means no data, why lots of grids in northwest and northeast are white? This paper is suggested for minor revisions.

---

## Referee Comment (RC2) · Anonymous Referee #2 · 5 Apr 2017

This is an important paper that worth publication. Author contribute the knowledge about the domestic transfer of the trade related air pollution and associated health impacts (calculated by the premature deaths). The methodology is mature and robust and the results are of great interests given the fact that China's air-pollution is currently the most timing issue. Several minor revision suggestions: 1) Definition of premature deaths, author should explain how the deaths are defined, my understanding is that the people are not directly "killed" by air pollution, it should be the shorten of expected life time, this need be clarify in the main text. 2) Given that only the domestic Input-output models are used in the analysis, author need to explain the impacts of international trade and related uncertainties. 3) Improve the policy implications in terms of how the trade policy can contribute to pollution mitigation.

---

## Author Comment (AC1) · 30 Jun 2017

**Referee #1**
This manuscript investigates the effect of atmospheric transport and trade on air pollution mortality in China. The topic is interesting and important. The approach is new and robust. The paper presents a good interdisciplinary study and is written. A few minor correction may need to be made.

**Response:** We thank Referee #1 for the encouragement and for the valuable comments to improve our manuscript. Responses to each point are addressed as below.

1. Abstract: "33% of China's air pollution deaths…", it is suggested to provide the number of premature deaths besides the percentage.

**Response:** Thanks for the suggestion. We have add the number there and some other regions.

2. $C_0$ is the counterfactual concentration in equation (1). The authors should declare the value of $C_0$ adopted in this study.

**Response:** Thanks for the suggestion. We have added the data for counterfactual concentration ($C_0$) and death incidence (B) in the appendix of the revised manuscript.

3. China also imports intermediate and finished goods from other regions. The authors should describe the calculation of international imports in methods.

**Response:** Thanks for the comments. Yes, as the most populous country, China plays an important role in international import as well as export. These would have substantial health impacts to other countries. In another recent paper published in Nature (Zhang et al., 2017), we estimated the cross-regional health impacts among 13 world regions (including China) from international trade and atmospheric transport perspectives, including health impact of China's import on other regions. In this work, we didn't repeat the analysis but added a citation to the study mentioned above.

4. Recent progress in transboundary air pollutants and its associated health effect should be mentioned in this study, some are listed below: Zhang Q, Jiang X, Tong D, et al. Transboundary health impacts of transported global air pollution and international trade[J]. Nature, 2017, 543(7647): 705-709. Li Y, Meng J, Liu J, et al. Interprovincial reliance for improving air quality in China: a case study on black carbon aerosol[J].Environmental science & technology, 2016, 50(7): 4118-4126.

**Response:** Thanks for the suggestion. The two papers and other related papers were cited in the revised manuscript.

5. The authors should thoroughly check the manuscripts to avoid grammar mistakes. For example, Page 6 Line 25, "which account for about 35%..." should be "which accounted for about 35%...", all "PM2.5" should be changed to "PM$_{2.5}$".

**Response:** Thanks for the comments. We have checked through the manuscript and corrected the grammar mistakes.

6. Fig.2, white color means no data, why lots of grids in northwest and northeast are white?

**Response:** Thanks for the comments. In Fig. 2, the grids where no death occurred were also labeled in white color. We have revised the Fig. 2 to distinguish the cases of "no data" and "no death".

**Reference:**
Zhang, Q., Jiang, X., Tong, D., Davis, S. J., Zhao, H., Geng, G., Feng, T., Zheng, B., Lu, Z., Streets, D. G., Ni, R., Brauer, M., van Donkelaar, A., Martin, R. V., Huo, H., Liu, Z., Pan, D., Kan, H., Yan, Y., Lin, J., He, K., and Guan, D.: Transboundary health impacts of transported global air pollution and international trade, Nature, 543, 705-709, 10.1038/nature21712, 2017.

---

## Author Comment (AC2) · 30 Jun 2017

**Referee #2**

This is an important paper that worth publication. Author contribute the knowledge about the domestic transfer of the trade related air pollution and associated health impacts (calculated by the premature deaths). The methodology is mature and robust and the results are of great interests given the fact that China's air-pollution is currently the most timing issue. Several minor revision suggestions:

**Response:** We thank Referee #2 for the positive comments. Responses to each point are addressed as below.

1. Definition of premature deaths, author should explain how the deaths are defined, my understanding is that the people are not directly "killed" by air pollution, it should be the shorten of expected life time, this need be clarify in the main text.

**Response:** Thanks for the comments. In this work, we estimated the premature mortality related to air pollution, that is, death of an individual before his or her life expectancy due to exposure to air pollution. We have clarified this in the revised manuscript.

2. Given that only the domestic Input-output models are used in the analysis, author need to explain the impacts of international trade and related uncertainties.

**Response:** Thanks for the comments. In the Sect. 3.4 of the revised manuscript, we discussed the limitations due to using the domestic MRIO.

3. Improve the policy implications in terms of how the trade policy can contribute to pollution mitigation.

**Response:** Thanks for the comments. We have strengthened the policy discussions in the Sect. 4 of the revised manuscript.